# Resveratrol Inhibits Oxidative Stress and Regulates M1/M2-Type Polarization of Microglia via Mediation of the Nrf2/Shh Signaling Cascade after OGD/R Injury In Vitro

**DOI:** 10.3390/jpm12122087

**Published:** 2022-12-19

**Authors:** Jie Liu, Hongyan Liao, Yue Chen, Huimin Zhu, Xuemei Li, Jing Liu, Qin Xiang, Fanling Zeng, Qin Yang

**Affiliations:** 1Department of Neurology, The First Affiliated Hospital of Chongqing Medical University, Chongqing 400016, China; 2Department of Neurology, Longevity District People’s Hospital of Chongqing, Chongqing 401220, China; 3Health Management Center, The First Affiliated Hospital of Chongqing Medical University, Chongqing 400016, China

**Keywords:** resveratrol, microglia, polarization, oxidative stress, Nrf2 signaling, Shh signaling, co-culture

## Abstract

Aims: Microglia are closely related to the occurrence and development of oxidative stress. Cerebral ischemia leads to abnormal activation of microglia. Resveratrol can regulate M1/M2-type microglia polarization, but the underlying mechanism is not well understood, although the Nrf2 and Shh signaling pathways may be involved. Given that resveratrol activates Shh, the present study examined whether this is mediated by Nrf2 signaling. Methods: N9 microglia were pretreated with drugs before oxygen-glucose deprivation/reoxygenation (OGD/R). HT22 neurons were also used for conditional co-culture with microglia. Cell viability was measured by CCK-8 assay. MDA levels and SOD activity in the supernatant were detected by TBA and WST-1, respectively. Immunofluorescence detected Nrf2 and Gli1 nuclear translocation. The levels of CD206, Arg1, iNOS, TNF-α, Nrf2, HO-1, NQO1, Shh, Ptc, Smo, Gli1 protein and mRNA were measured by Western blotting or RT-qPCR. Annexin V-FITC Flow Cytometric Analysis detected apoptosis. Results: Resveratrol and Nrf2 activator RTA-408 enhanced the viability of microglia, reduced oxidative stress, promoted M2-type microglia polarization and activated Nrf2 and Shh signaling. ML385, a selective inhibitor of Nrf2, decreased the viability of microglia, aggravated oxidative stress, promoted M1-type microglia polarization and inhibited Nrf2 and Shh signaling. Moreover, resveratrol and RTA-408-treated microglia can reduce the apoptosis and increase the viability of HT22 neurons, while ML385-treated microglia aggravated the apoptosis and weakened the viability of HT22 neurons. Conclusions: These results demonstrated that resveratrol may inhibit oxidative stress, regulate M1/M2-type polarization of microglia and decrease neuronal injury in conditional co-culture of neurons and microglia via the mediation of the Nrf2/Shh signaling cascade after OGD/R injury *in vitro*.

## 1. Introduction

Ischemic stroke is one of the three major life-threatening diseases with high incidence, high mortality, high disability rate and high recurrence rate [1,2]. Oxidative stress injury is one of the core pathological links after cerebral ischemia, and microglia are closely related to its occurrence and development [2,3]. In recent years, studies have shown that cerebral ischemia leads to abnormally activated microglia, which produce various free radicals and inflammatory factors, triggering oxidative stress, aggravating brain injury and hindering the recovery of neural function [3,4]. The activation process of microglia is the polarization process of M1 and M2-type microglia. M1-type microglia secrete pro-inflammatory factors, such as TNF-α, IL-1β and iNOS. M2-type microglia secrete anti-inflammatory factors, such as IL-10, IL-13, CD206 and Arg1. M1 and M2-type microglia can transform each other [2]. Therefore, there is a strong clinical transformation potential for the identification of drugs or methods that can regulate M1/M2-type polarization of microglia, thereby reducing oxidative stress injury and promoting the recovery of neural function after stroke. 

Resveratrol (trans-3, 5, 4′-trihydroxystilbene), a natural polyphenol compound, exists in grapes, peanuts, plums, red wines and other dietary sources, and has a variety of biological activities, including antioxidant, anti-inflammatory, anti-cancer, free radical scavenging and neuroprotective properties [5,6,7]. Moreover, resveratrol can promote M2-type microglia polarization [8]. However, it has not been fully elucidated how resveratrol regulates the polarization of microglia.

Nuclear factor E2-related factor 2 (Nrf2)/antioxidant response element (ARE) is the central regulator of cellular antioxidant response [9]. Nrf2, a cap’n’collar (CNC) transcription factor, regulates the intracellular redox balance and is expressed in all organs in the body. Under resting conditions, Nrf2 is located in the cytoplasm and binds to cytoplasmic protein partner Keap1 to maintain a continuous ubiquitination state. After the body undergoes an oxidative stress reaction, Keap1 is inactivated by direct modification of cysteine sulfhydryl residues, Nrf2 is released from Keap1, transported to the nucleus, binds to ARE, regulates the expression of multiphase II detoxification enzymes and anti-oxidation transcription activity of enzyme genes. Thereby, Nrf2 plays a vital role in regulating oxidative stress, inflammation, aging and ROS [9,10,11,12]. Heme oxygenase-1(HO-1) and NAD(P)H:quinone oxidoreductase 1 (NQO1) are two effective antioxidant and cytoprotective enzymes. They play a key role in protecting cells from oxidative stress and can be regulated by Nrf2 [11,13]. Previous studies *in vivo* have shown that resveratrol can upregulate the expression of Nrf2 and HO-1, and reduce ischemic brain damage [6,14]. It is unclear whether or how resveratrol regulates the polarization of microglia via Nrf2/HO-1 signaling.

The Sonic Hedgehog (Shh) signaling pathway includes Shh ligand, transmembrane protein receptors Patched (Ptc) and Smoothened (Smo), and downstream glioma-associated oncogene homolog (Gli) family transcription factors (Gli1, Gli2 and Gli3), and so on. Shh binds to the cell surface transmembrane protein receptor Ptc, Smo migrates to the primary cilia from the cytoplasm, then Gli transcription factors are activated, enter the nucleus, and initiate downstream effectors of the Shh pathway [15,16]. Previous studies have proved that Shh signaling plays an important role in cell proliferation, differentiation, migration and cell survival in the development of pathological conditions such as stroke and trauma [17]. Moreover, more studies have shown that resveratrol can activate the Shh signaling [7,18,19]. However, the role of the Shh signaling in the polarization of microglia after stroke is currently unknown. It is also unclear whether Nrf2 signaling affects Shh signaling during resveratrol regulation of the polarization of microglia.

Therefore, in the present study, N9 microglia were used as model cells to expound whether resveratrol affects oxidative stress and microglia polarization via the Nrf2/Shh signaling cascade after oxygen glucose deprivation/reoxygenation (OGD/R) injury *in vitro*. Further, an HT22 neuron and microglia co-culture model was used to test whether Nrf2 signaling mediated resveratrol to regulate neuronal apoptosis and viability. We definitively demonstrate that resveratrol and Nrf2 activator RTA-408 enhanced the viability of microglia, reduced oxidative stress, promoted M2-type microglia polarization and activated Nrf2 and Shh signaling. Nrf2 inhibitor ML385 decreased the viability of microglia, aggravated oxidative stress, promoted M1-type microglia polarization and inhibited Nrf2 and Shh signaling. Moreover, resveratrol and RTA-408-treated microglia can reduce the apoptosis and increase the viability of HT22 neurons, while ML385-treated microglia aggravated apoptosis and weakened the viability of HT22 neurons. In summary, resveratrol may inhibit oxidative stress, regulate M1/M2-type polarization of microglia and decreases neuronal injury in the conditional co-culture of neuron and microglia via the mediation of the Nrf2/Shh signaling cascade following OGD/R injury *in vitro*. This study indicates that resveratrol may have a new potential therapeutic target after cerebral ischemic injury.

## 2. Materials and Methods

### 2.1. Cell Culture 

N9 microglia (donated by Professor Xu Ying, Department of Anesthesiology, children’s Hospital Affiliated to Chongqing Medical University, Chongqing, China) were cultured in DMEM/F12 medium (Gibco; Thermo Fisher Scientific, Inc., Waltham, MA, USA) supplemented with 10% FBS (PAN, Adenbach, Germany) and HT22 neuron (donated by Dr. Tang Wei, Department of Neurosurgery, the First Affiliated Hospital of Chongqing Medical University, Chongqing, China) were cultured in DMEM medium (Gibco; Thermo Fisher Scientific, Inc., Waltham, MA, USA) supplemented with 10% FBS (PAN, Adenbach, Germany). All cultures were maintained at 37 °C in an incubator with a humidified atmosphere of 5% CO_2_. The cells were passaged at a ratio of 1:3 every 3 days.

### 2.2. OGD/R Model

An OGD/R model of N9 microglia was established to mimic cerebral artery occlusion and reperfusion injury, according to previously described methods with slight modifications [20]. In brief, following washing the cells three times with PBS solution, N9 microglia were cultured with serum-free medium and placed in an incubator with an anaerobic gas mixture containing 94% N_2_, 5% CO_2_, and 1% O_2_ (Thermo 3111; Thermo Fisher Scientific Inc., Waltham, MA, USA) at 37 °C for 3 h. During reoxygenation, the medium was re-placed by complete medium containing DMEM/F12 medium and 10% FBS, and the cells were maintained in an incubator with 95% air and 5% CO_2_ for 24 h. 

### 2.3. Establishment of Co-Culture Model by Condition Medium

In order to study the effect of resveratrol and other drugs-stimulated microglia on the survival of neurons, N9 microglia and HT22 neurons were co-cultured by an established conditioned medium method. According to the conditioned medium method, HT22 neurons (approximately 5 × 10^5^/mL) were seeded in poly-L-lysine-coated six-well plates and incubated for 2 days. N9 microglia (approximately 5 × 10^6^/mL) were seeded in poly-L-lysine-coated six-well plates and subjected to a variety of treatments as described above. The microglia supernatants were collected and centrifuged at 1000 rpm for 20 min, and then used as a conditioned medium to replace neuronal culture medium, and co-cultured for 24 h.

### 2.4. Drug Treatment 

To investigate whether resveratrol enhances microglial viability after OGD/R injury *in vitro*, four treatment groups were used for comparison: (1) Normal group (Nor), N9 microglial cells were cultured in complete medium without OGD/R; (2) Control group (Ctrl), N9 microglial cells were treated with OGD/R only; (3) vehicle group(Veh), N9 microglial cells were cultured in complete medium containing ethanol [1.3% v/v] for 24 h before OGD/R; (4) resveratrol pretreatment group (Res), N9 microglial cells were cultured in complete medium containing different concentrations [1, 5, 20, 40 and 80 uM] resveratrol for 24 h prior to OGD/R; and (5) blank group, complete medium without N9 microglia.

To determine whether resveratrol promotes microglia viability, alleviates oxidative damage of microglia, adjusts M1/M2 polarization of microglia and activates the Nrf2 signaling pathway after OGD/R, there were three groups: (1) Normal group (Nor), N9 microglial cells were cultured in complete medium without OGD/R; (2) Control group (Ctrl), N9 microglial cells were treated with OGD/R only; (3) 20 μM resveratrol pretreatment group (Res20), N9 microglial cells were maintained in complete medium containing 20 μM resveratrol for 24 h before OGD/R.

To explore whether Nrf2 signaling mediates resveratrol to enhance microglial viability, reduce oxidative stress, promote the M2 type polarized and impact Shh signaling after OGD/R injury, there were six groups: (1) Normal group (Nor), N9 microglial cells were cultured in complete medium without OGD/R; (2) Control group (Ctrl), N9 microglial cells were treated with OGD/R only; (3) 20 μM resveratrol pretreatment group (Res20), N9 microglial cells were maintained in complete medium containing 20 μM resveratrol for 24 h before OGD/R; (4) Nrf2 inhibitor 5 μM ML385 group, N9 microglial cells were maintained in complete medium containing 5 μM ML385 for 24 h before OGD/R; (5) resveratrol combined ML385 group (R + M), N9 microglial cells were maintained in complete medium containing 20 μM resveratrol and 5 μM ML385 for 24 h before OGD/R; (6) Nrf2 activator 100 nM RTA-408 group, N9 microglial cells were maintained in complete medium containing 100 nM RTA-408 for 24 h before OGD/R.

To examine whether resveratrol enhances HT22 neuronal viability, reduces neuronal apoptosis. and in-duces neurite outgrowth and synaptogenesis on the neuron-microglia condition co-culture via the Nrf2 signal, there are six groups: (1) normal group (Nor), HT22 neurons were cultured in culture medium of normal culture of N9 microglia for 24 h; (2) control group (Ctrl), HT22 neurons were cultured in culture medium of N9 microglia after OGD/R treatment for 24 h; (3) 20 μM resveratrol group (Res20), pretreat microglia with a medium containing 20 μM resveratrol for 24 h, and then culture the HT22 neurons for 24 h with OGD/R-treated microglial medium; (4) Nrf2 inhibitor 5 μM ML385 group, pretreat microglia with a medium containing 5 μM ML385 for 24 h, and then culture the HT22 neurons for 24 h with OGD/R-treated microglial medium; (5) resveratrol combined ML385 group (R + M), pretreat microglia with a medium containing 20 μM resveratrol and 5 μM ML385 for 24 h, and then culture the HT22 neurons for 24 h with OGD/R-treated microglial medium; (6) Nrf2 activator 100 nM RTA-408 group, pretreat microglia with a medium containing 100 nM RTA-408 for 24 h, and then HT22 neurons were cultured with OGD/R-treated microglial medium for 24 h.

### 2.5. Cell Viability Assay 

Cell viability was measured with Cell Counting Kit (CCK)-8 assay. Briefly, the N9 cells (approximately 6000/well) were seeded in poly-L-lysine-coated 96-well plates with six replicates per group, and subjected to a variety of treatments as described above. CCK-8 solution (10 uL/100 uL) was added to each culture well and cells were incubated for 2 h at 37 ℃. Absorbance at 450 nm was examined with a microplate reader (Thermo Labsystems, Vantaa, Finland). The experiment was repeated three times.

### 2.6. Measurements of SOD Activity and MDA Levels

The activity of SOD and the levels of MDA in N9 microglial supernatant were assessed using commercial assay kits (Nanjing Jiancheng Bioengineering Institute, Nanjing, China). SOD activity was based on the auto-oxidation of hydroxylamine and the developed blue color was measured at 550 nm. MDA levels were based on thiobarbituric acid (TBA) reactivity. In brief, after mixing trichloroacetic acid with the homogenate and centrifuging, a supernatant was obtained, and TBA was added. The developed red color of the resulting reaction was measured at 532 nm with a spectrophotometer. Other procedures were carried out following the manufacturer’s protocols. Each experiment was repeated three times.

### 2.7. Immunocytochemistry

N9 microglia were seeded on coverslips, subjected to various treatments as described above, then fixed with 4% paraformaldehyde for 10 min at room temperature, then washed three times with PBS and permeabilized with 0.3% Triton X-100 for 10 min at room temperature. After PBS washing three times, the cells were blocked with 10% normal goat serum for 45 min at 37 °C, then incubated at 4 °C overnight with the following primary antibodies: monoclonal rabbit anti-Nrf2 antibody (1:100; cat. no. ab62352; Abcam, Cambridge, UK), monoclonal mouse anti-CD206 antibody (1: 100, cat. no.60143-1-lg, Proteintech, Chicago, IL, USA), monoclonal mouse anti-iNOS antibody (1:100, cat.no.ab4999; Abcam, Cambridge, UK) and polyclonal rabbit anti-Gli1 antibody (1:100; cat.no.ab49314; Abcam, Cambridge, UK). After being washed with PBS, the cells were incubated with CY3-goat anti-rabbit or FITC-goat anti-mouse IgG (1: 100, Proteintech, Chicago, IL, USA) for 1 h at 37 °C. The primary antibodies were replaced with PBS in the negative controls. Nu-clei were counterstained with DAPI (Beyotime, Nantong, China) for 5 min in the dark. Finally, cells were determined with a laser confocal microscope (Zeiss, Jena, Germany). Each experiment was repeated three times.

### 2.8. Quantitative Real-Time PCR (RT-qPCR)

Total RNA was extracted by Trizol (TaKaRa, Dalian, China) according to the manufacturer’s instructions. Equal amounts of RNA were used to synthesize cDNA using the reverse transcription kit (TaKaRa, Dalian, China). PCR was carried out using the SYBR-Green RT-PCR kit (Qiagen). GAPDH was used as an internal control. RNA quantities of target genes were analyzed using the Ct method. The final results were normalized and are expressed as the fold change compared to the target gene/GAPDH. The primers used in the RT-PCR were as in Table 1: 

### 2.9. Western Blot Analysis 

After the cells were treated as above, N9 microglial cells were rinsed once with ice-cold PBS, lysed in RIPA lysis buffer containing 1% PMSF, and incubated on ice for 30 min followed by centrifugation at 12,000× *g* and 4 °C for 15 min. The protein concentration was observed by a BCA protein assay kit (Beyotime, Nantong, China). Protein samples were mixed with the 5× loading buffer at a ratio of 1:4 and boiled for 8 min, then resolved by 10% SDS-PAGE and transferred to PVDF membranes (Millipore, Billerica, MA, USA). The membranes were blocked with 5% defatted milk in TBST buffer for 2 h at room temperature and then incubated with primary antibodies overnight at 4 ℃. The primary antibodies used were: monoclonal mouse anti-CD206 antibody (1:1000, cat.no.60143-1-lg, Proteintech, Chicago, USA), monoclonal mouse anti-iNOS antibody (1: 1000, cat.no.ab4999; Abcam, Cambridge, UK), monoclonal rabbit anti-Nrf2 antibody (1:1000; cat.no.ab62352; Abcam, Cambridge, UK), monoclonal rabbit anti-HO-1 (1:1000; cat.no.ab68477; Abcam, Cambridge, UK), monoclonal rabbit anti-NQO1 (1:1000; cat.no.ab80588; Abcam, Cambridge, UK), monoclonal rabbit anti-Shh (1:1000; cat.no.ab68477; Abcam, Cambridge, UK), polyclonal rabbit anti-Ptc-1 (1:1000; cat.no.ab53715; Abcam, Cambridge, UK), polyclonal rabbit anti-Smo (1:1000; cat.no.ab236465; Abcam, Cambridge, UK), polyclonal rabbit anti-Gli1 antibody (1:100; cat.no.ab49314; Abcam, Cambridge, UK)rabbit, and anti-β-actin (1:500; cat.no.GB1101; Servicebio, Wuhan, China), monoclonal rabbit anti-GAPDH (1:500; cat.no. AB-P-R001; Goodhere, Hangzhou, China) and monoclonal mouse anti-GAPDH (1:500; cat.no. AB-M-M001; Goodhere, Hangzhou, China) as loading controls. The membranes were washed with TBST buffer and incubated with the second antibody peroxidase-conjugated affiniPure goat anti-rabbit IgG (1:5000; cat.no.ZB-2301; ZSGB-BIO, Beijing, China) and goat anti-mouse IgG (1:1000; cat.no.A0216; Beyotime, Nantong, China) at 37 ℃ for 1 h. The target bands were detected by the enhanced chemiluminescence method and the images were analyzed with Quantity One software (Bio-Rad, Hercules, CA, USA). The gray ratio of the target protein to β-actin represented the expression level of the target protein. Experiments were repeated three times.

### 2.10. Annexin V-FITC Flow Cytometric Analysis of HT22 Neuronal Apoptosis 

HT22 neurons under the different culture conditions were collected simultaneously. The cells were dissociated into single cells, centrifuged with PBS buffer at 1000 r/min for 5 min, resuspended in 500 uL PBS, filtered once on a 400 mesh screen, centrifuged at 500 to 1000 r/min for 5 min, PBS removed, stained with 1 mL of PI staining solution, placed in the dark at 4 °C for 30 min, and then detected with the cell flow analysis. The data were analyzed using FlowJo software (CytExpert 2.0). Each experiment was repeated three times.

### 2.11. Statistical Analysis

Quantitative data are expressed as the mean ± standard deviation and the results are presented from three independent experiments. Data were statistically analyzed by a one-way analysis of variance followed by the Student t-test for single comparisons and Tukey’s post hoc test for multiple comparisons. All data were analyzed using SPSS v.22.0 for Windows. *p* < 0.05 was considered to be statistically significant difference.

## 3. Results

### 3.1. Concentration Effect of Resveratrol on Viability of N9 Microglia Following OGD/R Injury 

In order to investigate the effect of resveratrol pretreatment on the antioxidation and polarization of microglia following OGD/R injury, we compared cell viability with different concentrations of resveratrol with the CCK-8 assay. As Figure 1 shows, that resveratrol pretreatment (1, 5, 20, 40 and 80 μM) significantly reduced oxygen glucose deprivation/reoxygenation (OGD/R)-induced cytotoxicity in a concentration-dependent manner in N9 microglial cells. As the most potent concentration was 20 μM, we selected 20 μM resveratrol in the following studies.

### 3.2. Resveratrol Pretreatment Ameliorates Oxidative Damage and Regulates M1/M2 Polarization of N9 Microglia Following OGD/R Injury In Vitro

Oxidative stress is accepted to play a major role in the neurodegenerative process after Stroke [21,22,23]. To investigate the effect of resveratrol on the oxidative stress in OGD/R-induced N9 cells, the markers of oxidative stress including SOD and MDA were measured. As shown in Figure 2A,B, oxygen glucose deprivation/reoxygenation (OGD/R)-induced N9 microglia decreased SOD activity and increased MDA levels, while resveratrol pretreatment had the opposite effect, with significantly increased SOD activity and decreased MDA levels. These results indicated that resveratrol pretreatment attenuated oxidative stress of N9 microglia following OGD/R injury *in vitro*.

Morphological changes in microglia cannot always accurately reflect the activation status. CD206 and Arg1 are the M2 phenotype microglia markers, iNOS and TNF-α are the M1 phenotype microglia markers [20,24]. Therefore, Western blotting and RT-PCR assay were used to explore the protein and mRNA levels of CD206, Arg1, iNOS and TNF-α. We found that the protein and mRNA expression of M1 markers (iNOS and TNF-α) were significantly increased in only OGD/R-treated N9 cells, while resveratrol could suppress M1 marker expression and significantly promote microglia polarization to the M2 phenotype in conditions of inflammatory injury, as indicated by the increased expression of M2 markers (CD206, Arg1) (Figure 2C–I). These results showed that resveratrol pretreatment accelerated microglial polarization to the M2 phenotype following OGD/R injury *in vitro*.

### 3.3. Resveratrol Pretreatment Upregulates Expression of Nrf2, HO 1 and NQO 1 Proteins in N9 Microglia Following OGD/R Injury In Vitro

To investigate whether resveratrol affects microglia through activation of the Nrf2 signaling pathway, we examined the levels of Nrf2 and the related signaling pathways protein with immunofluorescence and Western blotting assay. Firstly, immunofluorescence showed that the Nrf2 protein was mainly located in cytoplasm of microglia in the resting condition (Figure 3F,M), after only OGD/R, few cells were positive for Nrf2 in the nucleus (Figure 3I,M), while the Nrf2 protein of resveratrol pretreated microglia mainly existed in the nucleus (Figure 3L,M). Next, as shown in Figure 3N–Q, there was a slight increase in the protein expression levels of Nrf2, HO-1, and NQO1 post OGD/R damage, while resveratrol pretreatment enhanced OGD/R -induced protein expression levels of Nrf2, HO-1, and NQO1. These results indicated that resveratrol promoted Nrf2 protein transfer into the nucleus, and upregulated expression of Nrf2, HO-1 and NQO1 proteins in N9 microglia following OGD/R injury *in vitro*. In other words, resveratrol enhanced activation of the Nrf2 signaling in N9 microglia after OGD/R injury.

### 3.4. Nrf2 Signaling Mediates the Effects of Resveratrol to Inhibit Oxidative Stress and Regulate M1/M2 Phenotype Polarization of Microglia Following OGD/R Injury In Vitro

In the present study, we used Nrf2 inhibitor ML385 and Nrf2 activator RTA-408 to investigate the role of Nrf2 signaling in resveratrol-ameliorating oxidative stress and resveratrol-regulating M1/M2 phenotype polarization of N9 microglia following OGD/R injury *in vitro*.

First, we detected whether resveratrol, ML385 and RTA-408 affected oxidative stress of N9 microglia after OGD/R injury with SOD activity and MDA levels. As Figure 4A shows, OGD/ R showed an inhibitory effect on the activity of SOD in N9 cells, and ML385 pretreatment slightly strengthened its effect. However, resveratrol or RTA-408 pretreatment increased the activity of SOD. ML385 also blocked the ameliorative effect of resveratrol on the activity of SOD. Conversely, OGD/R induced the production of MDA, ML385 pretreatment also amplified its effect, while the induction effect was attenuated by resveratrol or RTA-408 pretreatment. Similarly, ML385 blocked the depressed effect of resveratrol on the MDA levels (Figure 4B). These results indicate that Nrf2 signaling mediates resveratrol to inhibit oxidative stress of N9 microglia after OGD/R injury. 

Next, we investigated whether resveratrol, ML385 and RTA-408 affected M1/M2 phenotype polarization of N9 microglia after OGD/R injury with immunofluorescence and RT-PCR. As Figure 4C–R show, resveratrol or RTA-408 pretreatment on N9 microglia after OGD/R injury could significantly promote the protein and mRNA expression of M2 markers (CD206, Arg1), and suppress M1 marker expression (iNOS, TNF-α), while, ML385 pretreatment could decrease M2 marker expression (CD206, Arg1) and increase M1 marker expression (iNOS, TNF-α), it also weakened resveratrol from microglia polarization to the M2 phenotype. These results demonstrate that Nrf2 signaling mediates resveratrol to regulate M1/M2 phenotype polarization of microglia following OGD/R injury *in vitro*.

### 3.5. Nrf2 Signaling Mediates Resveratrol to Affect Shh Signaling Pathway in N9 Microglia Following OGD/R Injury

Our previous studies showed that resveratrol can upregulate Nrf2 and Shh signaling, respectively [7,14,17,25]. In addition, the canonical Shh pathway may play a role through the Nrf2/Shh signaling axis in liver tumors [26]. However, it remains unclear whether Nrf2 signaling mediates resveratrol to affect Shh signaling in the brain.

Immunofluorescence assay showed that the Gli1 protein was mainly located in cytoplasm of microglia in the resting condition (Figure 5B,P). As for ML385 pretreatment or only OGD/R treatment, few cells were positive for Nrf2 in the nucleus (Figure 5C,Q). Meanwhile, the Gli1 protein of resveratrol- or RTA-408-prereated microglia mainly existed in the nucleus (Figure 5D,R,G,U). In addition, there was no significant difference in the expression level of Gli1 protein in the cytoplasm and nucleus of the resveratrol combined with ML385 pretreatment (Figure 5F,T). Moreover, Western blotting showed that resveratrol or RTA-408 pretreatment significantly increased the expressions of Shh, Ptc, Smo and Gli1 protein, and there was no significant difference between them. There was a slight increase in the protein expression levels after only OGD/R damage, whereas ML385 mitigated its effects, and attenuated the positive effects of resveratrol in activating the Shh signaling pathway (Figure 5V–Z). These results showed that Nrf2 signaling can mediate resveratrol to affect Shh signaling following OGD/R injury in the microglia.

### 3.6. Nrf2 Signaling Mediated Resveratrol to Regulate Neuronal Apoptosis and Viability in Neuron-Microglia Co-Culture

To investigate the effect of N9 microglial-conditioned medium on the neuron cells, the apoptosis and viability were measured by Annexin V-FITC/PI double staining and CCK-8 assay. As shown in Figure 6A,G, neuronal apoptosis was lowest in untreated microglia-neuron co-culture. There was a significant increase in apoptosis of HT22 cells that co-cultured with N9 microglia pretreated with or without ML385 before OGD/R injury, especially with ML385 (Figure 6B,D,G). While, the resveratrol- or RTA-408-treated co-culture system can reduce neuronal cell apoptosis (Figure 6C,F,G), and resveratrol could reduce the effect of ML385 on neuronal cell apoptosis (Figure 6E,G). Further, as shown in Figure 6H, the viability of HT22 neurons co-cultured with OGD/R-induced N9 microglia was reduced. The ML385 pretreated co-culture system also significantly reduced neuronal activity. However, the resveratrol- or RTA-408-treated co-culture system could enhance neuronal cell viability, but ML385 could weaken the ameliorative effect of resveratrol. To sum up, resveratrol- and Nrf2-activator RTA-408-treated microglia can reduce the apoptosis and increase the viability of HT22 neuronal cells, while Nrf2-inhibitor ML385-treated microglia aggravated apoptosis and weakened the vitality of HT22 neuronal cells.

## 4. Discussion

The present study showed that resveratrol pretreatment increased viability, inhibited oxidative stress, and promoted microglia polarization to M2 phenotype following microglia OGD/R injury *in vitro*. Moreover, resveratrol reduced apoptosis and increased the viability of HT22 neurons following neuron and microglia co-culturing. These effects were mediated via activation of Nrf2 signaling and were abolished by treatment with ML385, the Nrf2 inhibitor. Conversely, activation of Nrf2 signaling by RTA-408, the Nrf2 activator, produced effects similar to resveratrol. In addition, resveratrol and RTA-408 increased, whereas ML385 decreased, the expression of Shh, Ptc, Smo, and Gli-1 protein. Taken together, these findings provide the first evidence that resveratrol inhibits oxidative stress, regulates M1/M2-type polarization of microglia and decreases neuronal injury in the conditional co-culture of neuron and microglia via the mediation of the Nrf2/Shh signaling cascade following OGD/R injury *in vitro*.

It was reported that 1, 12.5 and 25 μM resveratrol inhibited inflammation of N9 microglia induced by Aβ 1-42, but 50 μM and 100 μM resveratrol had cytotoxicity to microglia [27]. In LPS-activated BV2 microglia, 10 μg/mL resveratrol protected against LPS toxicity, 50 μM resveratrol attenuated pro-inflammatory gene expression and 100 ug/mL resveratrol decreased NO production, but did not protect against LPS toxicity [28,29]. Furthermore, 60 μM resveratrol ameliorated LPS-induced rat primary microglial activation [30]. Our present study showed that 5, 20 and 40 μM resveratrol enhanced the viability of N9 microglia after OGD/R injury. These studies indicate that the effects of resveratrol on microglia depend on the microglial cell line, induced model, or resveratrol preservation state.

Oxidative stress, an imbalance between oxidative and anti-oxidative systems of cells and tissues, is a result of excessive production of oxidative-free radicals and related reactive oxygen species (ROS) [31]. In recent years, studies have shown that the one of the core pathological links of OGD/R injury is the consequence of severe oxidative stress reaction caused by the disorder of free radical chain reactions [21,22,23]. SOD, an important free radical scavenger, plays a vital role in the oxidation and antioxidant balance, which can protect organisms from free radical attacks. The level of SOD activity indirectly reflects the ability to get rid of oxygen free radicals [32,33,34]. MDA, a product of lipid peroxidation, is one of the markers of oxidative stress [35]. Studies showed that the level of MDA is indirectly reflected in the severity of the free radical attack on cells or organisms and is often used as the degree of lipid peroxidation [36,37]. In our present study, SOD activity was significantly lower and MDA levels were higher in the control group than that in the normal group. The results indicated that OGD/R induced microglial oxidative damage. Resveratrol or Nrf2 activator RTA-408 pretreatment significantly increased the activity of SOD and reduced the levels of MDA, and Nrf2 inhibitor ML385 pretreatment reversed the above results following OGD/R-induced micro-glia injury. Previously, a study reported that resveratrol attenuated oxidative stress of neural stem cells after OGD/R by upregulating the expression of Nrf2, HO-1 and NQO1 *in vitro* [25]. Here, our study further indicated that resveratrol can also inhibit oxidative stress damage of N9 microglia following OGD/R injury by activating Nrf2 signaling *in vitro*. 

Microglia are the innate immune cells of the central nervous system and play a defensive role in maintaining the microenvironment [38]. Under physiological conditions, microglia are at rest, but under pathological conditions, they are activated [39]. The activated microglia, with distinct phenotypic changes to the harmful M1 and neuroprotective M2 types, become a double-edged sword after stroke [40]. Therefore, it is more meaningful to explore the direction of microglial polarization than microglial activation. Chen et al. reported that IMM-H004 modulated BV2 microglia polarization to shift from M1 to M2 phenotype after OGD/R injury *in vitro* [41]. Yang et al. suggested that resveratrol promoted microglia polarization to the M2 phenotype in LPS-induced BV2 microglia injury via PGC-1α [8]. Wang et al. stated that HP-1c suppressed microglial polarization to the M1 phenotype and reduced oxidative stress of LPS-induced mouse primary microglia with the AMPK-Nrf2 pathway [42]. Here, our study showed that resveratrol and Nrf2 activator RTA-408 pretreatment promoted microglia polarization to M2 type, and Nrf2 inhibitor ML385 pretreatment led to the consequence of polarization toward M1 type following OGD/R-induced N9 microglia injury. Therefore, these results indicated that M1/M2 phenotype polarization of microglia may be mediated through multiple signaling pathways.

Nrf2 is an important transcription factor in the induction of various antioxidants, which regulates the cellular antioxidant response against oxidative stress injury. Resveratrol has an anti-oxidant ability through activating Nrf2 signaling in multiple organ injuries, such as heart, liver, kidney and brain [43,44,45,46]. After OGD/R injury *in vitro* or cerebral ischemia *in vivo*, resveratrol can attenuate neuronal apoptosis and cerebral ischemic injury, and promote proliferation of neural stem cells via enhancing the activation of the Nrf2 signaling pathway [14,22]. In addition, resveratrol can also activate the Shh signaling pathway, which not only has the effects of anti-oxidation, anti-inflammatory and anti-apoptosis, but also promotes neurogenesis, axon remodeling and the recovery of neurologic function [6,7,17,47,48,49,50,51]. In hepatocellular carcinoma, Nrf2 can promote tumor-initiating cell lineage and drug resistance by upregulating Shh expression [23]. In pancreatic stellate cells, Cav-1 ablation enhanced the growth of pancreatic cancer via Nrf2-induced Shh signaling [52]. However, in HNSCC cells, the knockdown of Nrf2 didn’t change the expression of Shh and Gli1 protein [53]. Here, our research indicated that resveratrol and the Nrf2 activator RTA-408 both activated the Shh signaling pathway, while the Nrf2 inhibitor ML385 downregulated Shh signaling in N9 microglia following OGD/R. Therefore, Shh signaling may be downstream of the Nrf2 signal in the brain.

Microglia have been considered as active contributors to neuronal damage in stroke, where excessive activation and abnormal regulation of microglia might lead to catastrophic and progressive neurotoxic con-sequences [39]. Activated microglia release protective and cytotoxic factors, which, in turn, affect the survival and function of peripheral neurons [54]. When this process regulates imbalances, reactive microglial cells cause neuroinflammatory damage by generating ROS and pro-inflammatory cytokines [38]. Therefore, it is meaningful to construct a microglia-neuron co-culture model to study the microenvironment of the brain *in vivo*. Moreover, co-culture models, including direct contact and conditioned medium co-culture models, can also get rid of the limitations and uncertainties brought about by single-cell culture. Wang et al. have shown that HP-1c reduced microglia-induced neuronal injury via CaMKKβ-dependent AMPK activation by a primary microglia-N1E-115 neuroblasts conditional culture model [42]. Lu et al. used the co-culture of primary neurons and microglia, the results showed that microglia promoted the expression and release of pro-inflammatory factors, thereby aggravating neuronal apoptosis via the TLR4/MyD88/NF-κB signaling pathway [55]. In the present study, resveratrol and Nrf2 activator RTA-408 pretreatment reduced microglia-induced neuronal apoptosis and enhanced neuronal activity and Nrf2 inhibitor ML385 pretreatment reversed the above results following N9 microglia-HT22 hippocampal neurons co-culture. Therefore, we came to a possible conclusion that resveratrol can reduce microglia-induced neuronal injury through regulation of the Nrf2 signaling pathway.

Taken together, these findings highlight resveratrol as a potential therapeutic candidate for regulating M1/M2 type polarization of microglia and ameliorating oxidative stress injury during OGD/R injury. This study suggests a new strategy to target the Nrf2/Shh signaling pathways in cerebral ischemic disorders. However, further studies will be needed to evaluate the mechanism of the role of resveratrol in cerebral ischemic diseases.

## Figures and Tables

**Figure 1 jpm-12-02087-f001:**
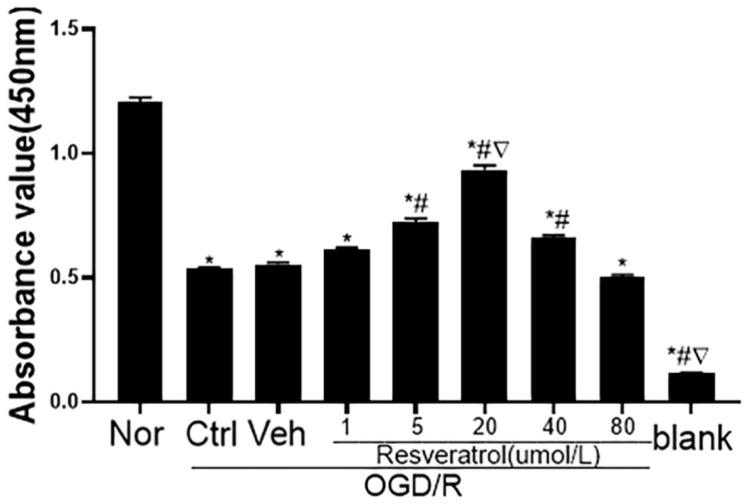
Concentration effect of resveratrol on the viability of N9 microglia following OGD/R injury. The viability of N9 microglia was detected with the CCK-8 assay. * *p* < 0.05, vs. Nor group; # *p* < 0.05, vs. Ctrl group; ▽ *p* < 0.05, vs. Res5 and Res40 groups. Data are presented as the mean ± standard deviation and were compared using a one-way analysis of variance (*n* = 3 for each group). OGD/R, oxygen-glucose deprivation/reoxygenation; Nor, normal; Ctrl, control; Veh, vehicle; Res, resveratrol.

**Figure 2 jpm-12-02087-f002:**
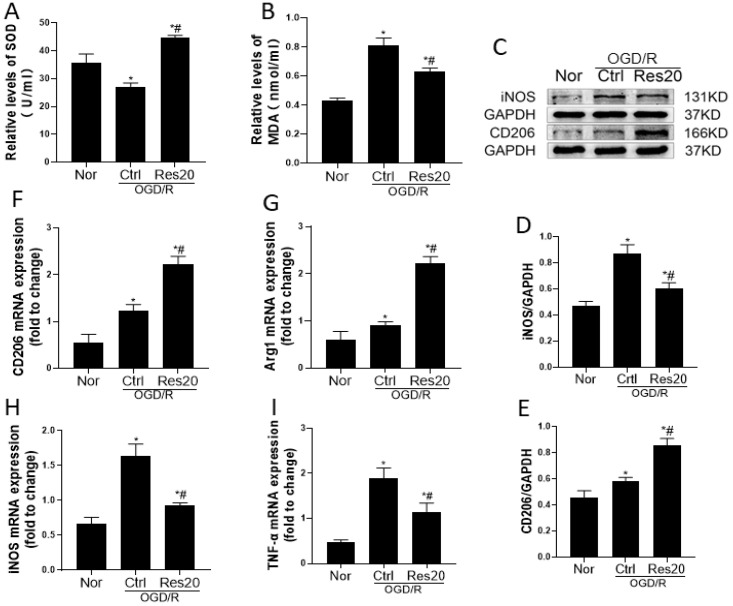
Resveratrol pretreatment ameliorates oxidative damage and regulates M1/M2 polarization of N9 microglia following OGD/R injury *in vitro*. (**A**) SOD activity detected by WST-1 assay. (**B**) MDA level detected by TBA assay. (**C**) Protein expressions of iNOS and CD206 with Western blot analysis. (**D,E**) Quantification of data for iNOS and CD206 proteins. (**F**–**I**) The levels of CD206, Arg1, iNOS and TNF-α mRNA were detected by RT-PCR. * *p* < 0.05, vs. Nor group; # *p* < 0.05, vs. Ctrl group. Data are presented as the mean ± standard deviation and were compared using a one-way analysis of variance (*n* = 3 for each group). OGD/R, oxygen-glucose deprivation/reoxygenation; Nor, normal; Ctrl, control; Res20, 20 μM resveratrol.

**Figure 3 jpm-12-02087-f003:**
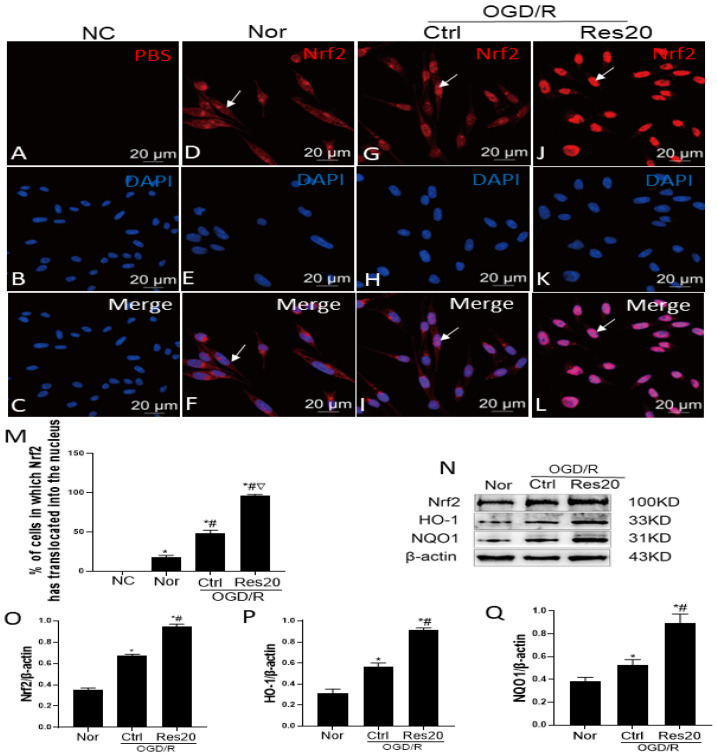
Resveratrol strengthens activation of the Nrf2 signaling pathway in N9 microglia following OGD/R injury *in vitro*. (**D,G,J**) Microglia were immunostained with antibodies to Nrf2 (red). (**B,E,H,K**) Nuclei were labeled with DAPI (blue). (**A**) The Nrf2 antibodies were replaced with PBS to serve as a negative control (NC). (**C,F,I,L**) Merges of A/B, D/E, G/H, J/K; Scale bar = 20 um. (**J**) Protein expressions of Nrf2, HO-1, and NQO1 with western blot analysis. (**L–M**) Quantification of data for Nrf2, HO-1 and NQO1 proteins. * *p* < 0.05, vs. NC group; # *p* < 0.05, vs. Nor group; ▽ *p* < 0.05, vs. Ctrl group. Data are presented as the mean ± standard deviation and were compared using a one-way analysis of variance (*n* = 3 for each group). Nor, normal; Ctrl, control; Res20, 20 μM resveratrol.

**Figure 4 jpm-12-02087-f004:**
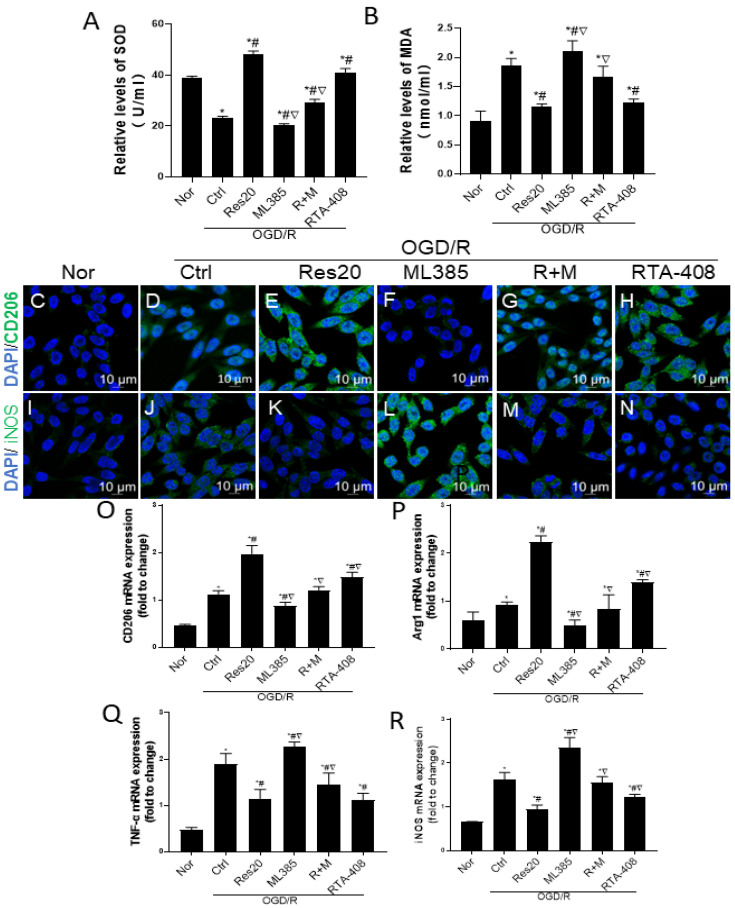
Nrf2 signaling mediates the effects of resveratrol to inhibit oxidative stress and regulate M1/M2 phenotype polarization of microglia following OGD/R injury *in vitro*. (**A**) SOD activity detected by WST-1 assay. (**B**) MDA level detected by TBA assay. (**C**–**H**): merges of immunofluorescence staining CD206 (green) and DAPI (blue); (**I**–**N**): merges of immunofluorescence staining iNOS (green) and DAPI (blue); Scale bars =10 um. (**O**–**R**): the expression levels of CD206, Arg1, iNOS and TNF-α mRNA with RT-PCR; * *p* < 0.05 vs. Nor; # *p* < 0.05 vs. Ctrl; ▽ *p* < 0.05 vs. Res20. Data are presented as the mean ± standard deviation and were compared using a one-way analysis of variance (*n* = 3 for each group). Nor, normal; Ctrl, control; Res20, 20 μM resveratrol, ML385, Nrf2 inhibitor; R+M, 20 μM resveratrol combination with ML385.RTA-408, Nrf2 agonist.

**Figure 5 jpm-12-02087-f005:**
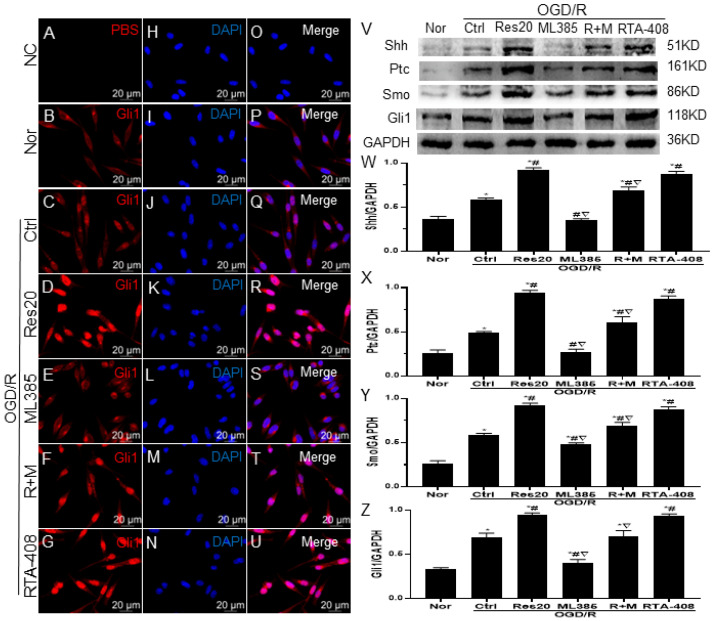
Nrf2 signaling mediates resveratrol to affect Shh signaling pathway in microglia. (**A**–**G**): Microglia were immunostained with antibodies to Gli1 (red). (**H**–**N**): Nuclei were labeled with DAPI (blue). (**O**–**U**): merges of A/H, B/I, C/J, D/K, E/L, F/M and G/N; Scale bars =20 um. (**V**): the expression of Shh, Ptc, Smo and Gli1 protein with Western blot. (W-Z): Quantitative analysis of Shh, Ptc, Smo and Gli1 protein levels. * *p* < 0.05 vs. Nor; # *p* < 0.05 vs. Ctrl; ▽ *p* < 0.05 vs. Res20. Data are presented as the mean ± standard deviation and were compared using a one-way analysis of variance (*n* = 3 for each group). Nor, normal; Ctrl, control; Res20, 20 μM resveratrol, ML385, Nrf2 inhibitor; R + M, 20 μM resveratrol combination with ML385; RTA-408, Nrf2 agonist.

**Figure 6 jpm-12-02087-f006:**
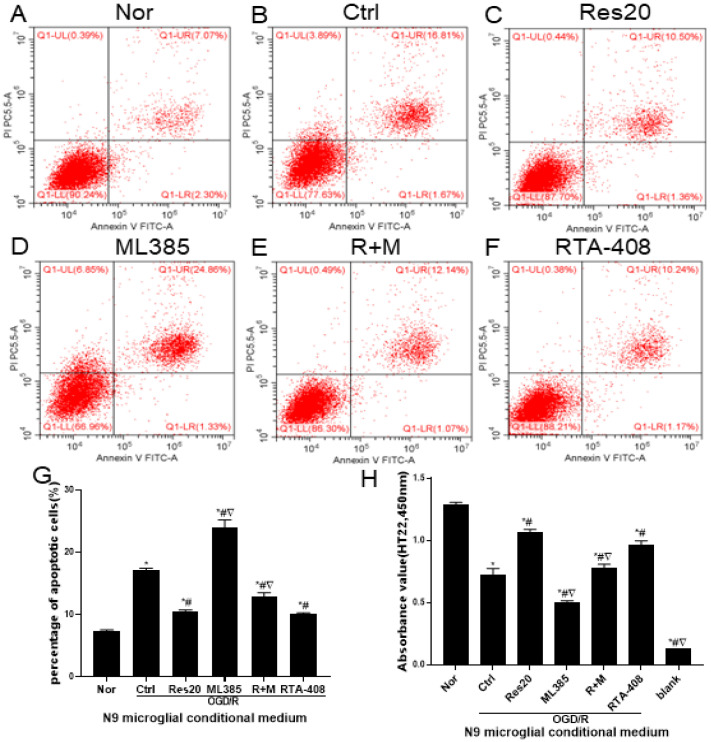
Nrf2 signaling mediated resveratrol to regulate neuronal apoptosis and viability in neuron-microglia co-culture. (**A**–**F**) Neuronal apoptosis detected by the Annexin V-FITC flow cytometer. (**G**) Percentage of apoptotic cells. (**H**) Neuronal viability detected by CCK-8 assay. * *p* < 0.05 vs. Nor; # *p* < 0.05 vs. Ctrl; ▽ *p* < 0.05 vs. Res20. Data are presented as the mean ± standard deviation and were compared using a one-way analysis of variance (*n* = 3 for each group). Nor, normal; Ctrl, control; Res20, 20 μM resveratrol, ML385, Nrf2 inhibitor; R + M, 20 μM resveratrol combination with ML385; RTA-408, Nrf2 agonist.

**Table 1 jpm-12-02087-t001:** Primer sequence.

Genes	Forward	Reverse
CD206	5′-GTCAACCCAAGGGCTCTTCTAA-3′	5′-AGGTGGCCTCTTGAGGTATGTG-3′
Arg1	5′-GGAACTCAACGGGAGGGTAAC-3	5′-GAAGGCGTTTGCTTAGTTCTGTC-3′
iNOS	5′-TTGGCTCCAGCATGTACCCT-3′	5′-TCCTGCCCACTGAGTTCGTC-3′
TNF-α	5′-CCAACGGCAGGATCTCAAAG-3′	5′-TGACGGTGTGGGTGAGGAGC-3′
GAPDH	5′-GACATCAAGAAGGTGGTGAAGC-3′	5′-GAAGGTGGAAGAGTGGGAGTT-3′

## Data Availability

Not applicable.

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
