# Peer review of "Resveratrol Inhibits Oxidative Stress and Regulates M1/M2-Type Polarization of Microglia via Mediation of the Nrf2/Shh Signaling Cascade after OGD/R Injury In Vitro"

_jpm, 2022, doi:10.3390/jpm12122087_

Round 1
Reviewer 1 Report
- The font size should be similar for all paragraphs
- It is important to give some images, demonstrations and illustrations to the methods used
Author Response
Response to Reviewer 1 Comments
Point 1: The font size should be similar for all paragraphs.
Response 1: Thank you for your good suggestion. The font size has been modified in the paper. Please check it.
Point 2: It is important to give some images, demonstrations and illustrations to the methods used.
Response 2: Thank you for your good suggestions. Our method is universal and has been proven repeatedly [1-5]. Therefore, we don't think he is necessary to give some images, demonstrations and illustrations to the methods used.
Reference
- Ren J, Fan C, Chen N, et al. Resveratrol pretreatment attenuates cerebral ischemic injury by upregulating expression of transcription factor Nrf2 and HO-1 in rats. Neurochem Res. 2011; 36(12):2352-62.
- Yu PP, Wang L, Tang F, et al. Resveratrol pretreatment decreases ischemic injury and improves neuro-logical function via sonic hedgehog signaling after stroke in rats. Mol Neurobiol. 2017;54( 1) : 212-226.
- Tang F, Guo S, Liao H, et al. Resveratrol enhances neurite outgrowth and synaptogenesis via sonic hedgehog signaling following oxygen-glucose deprivation/reoxygenation injury. Cell Physiol Biochem. 2017; 43( 2) : 852-869.
- Guo S, Liao H, Liu J, et al. Resveratrol Activated Sonic Hedgehog Signaling to Enhance Viability of NIH3T3 Cells in Vitro via Regulation of Sirt1. Cell Physiol Biochem. 2018;50(4):1346-1360.
- Okorji UP, Velagapudi R, El-Bakoush A, Fiebich BL, Olajide OA. Antimalarial Drug Artemether Inhibits Neuroinflammation in BV2 Microglia Through Nrf2-Dependent Mechanisms. Mol Neurobiol. 2016 Nov;53(9):6426-6443).

Reviewer 2 Report
Dear Authors,
Congratulations! the manuscript reads well. The authors have well presented the results and have stated the conclusion that resveratrol inhibits oxidative stress and mediates immune polarisation via Nrf2/Shh pathway. There are a few changes in phrasing suggested in the comments in the pdf attachment.

Author Response
Response to Reviewer 2 Comments
Point 1: Congratulations! the manuscript reads well. The authors have well presented the results and have stated the conclusion that resveratrol inhibits oxidative stress and mediates immune polarisation via Nrf2/Shh pathway. There are a few changes in phrasing suggested in the comments in the pdf attachment.
Response 1: Thank you very much for your recognition and suggestion. We have revised the draft according to your suggestion. Please check it.
- The statements of “Ischemic stroke is one of the three major diseases for threatening human health and life. It has the characteristics of high incidence, high mortality, high disability rate and high recurrence rate [1,2]." were corrected as“Ischemic stroke is one of the three major life threatening diseases with high incidence, high mortality, high disability rate and high recurrence rate [1,2]”.
- The statements of “Oxidative stress injury is one of the core pathological links after cerebral ischemia, and microglia is closely related to the occurrence and development of oxidative stress [2,3]." were corrected as “Oxidative stress injury is one of the core pathological links after cerebral ischemia, and microglia is closely related to its occurrence and development [2,3]”.
- The statements of “In recent years, studies have shown that cerebral ischemia leads to abnormal activation of microglia. The abnormally activated microglia produce various free radicals and inflammatory factors," were corrected as “In recent years, studies have shown that cerebral ischemia leads to abnormally activated microglia, which produce various free radicals and inflammatory factors,”.

Reviewer 3 Report
The manuscript by Liu J. et al. entitled 'Resveratrol inhibits oxidative stress and regulates M1/M2-type polarisation of microglia via the mediation of the Nrf2/Shh signaling cascade after OGD/R injury in vitro' investigates the ability of resveratrol, a known antioxidant molecule, to activate the Nrf2/Shh pathway and how this signal is able to promote an anti-inflammatory state in an immortalized murine microglia cell line subjected to OGD/R injury. Furthermore, the authors investigate the effect of OGD/R microglia on neuronal cultures, here represented by HT22 immortalized cells.
Although the work appears well structured, I consider the manuscript inappropriate for publication in a journal dealing with personalized medicine. I therefore advise the authors to proceed with submission to another journal.
I also believe that major revisions are necessary.
- First of all, I think one of the biggest limitations of this work lies in the known very poor bioavailability of resveratrol, so the possibility of a direct effect of resveratrol on microglia must be excluded. Furthermore, the effects of resveratrol on all cell types are undoubted so, still thinking of an in vivo model, the beneficial effect of resveratrol can only be understood as a peripheral effect (e.g. gut microbiota) that is mirrored to a central one. I believe that the authors should approach this topic objectively and if possible conduct pilot experiments in vivo to demonstrate the activation of this pathway in the microglia following resveratrol administration. As it stands, the manuscript is of little interest and I would advise the authors to stress rather on how activation or inhibition of the Nrf2/Shh pathway via ligands is able to prevent or exacerbate OGD/R/ischaemic stroke damage, and not on resveratrol.
- How does the effect of resveratrol, ML385, RTA-408 influence the M1/M2 balance of microglia in the absence of OGD/R damage?
- In figure 2, I advise the authors to also report the expression (mRNA) of arginase1 and TNF-a. Furthermore, I believe that quantifying pro/anti-inflammatory cytokines in the conditioned medium of microglia is necessary for all experimental groups.
- In figure 3, the authors should show a graph with the % of cells in which Nrf2 has translocated into the nucleus and not just the pictures.
- The authors demonstrate the effects of the conditioned medium of microglia on HT22 cells by taking them as a neuronal model. HT22 cells are not neurons unless they undergo a differentiation protocol. The authors did not provide any information about possible differentiation protocols. The authors must provide images of HT22 stained for neuronal markers.
Changes needed in the text:
- In the "drug treatment" section, the abbreviation "OGD" is often given instead of "OGD/R". Is this a typo or intentional?
- Table 1 is missing the sequences of the GAPDH primers
- For a better understanding of the result, I consider it necessary to add the words "OGD/R" to each graph.
- Improper use of the word "group" and coordinating conjunctions (e.g. and). The presentation of the results must be improved.
Round 2
Reviewer 3 Report
Dear Authors,
the manuscript has been improved following the suggestions given.
Sincerely,
Best Regards